# Habitual Aerobic Exercise Diminishes the Effects of Sarcopenia in Senescence-Accelerated Mice Prone8 Model

**DOI:** 10.3390/geriatrics5030048

**Published:** 2020-09-09

**Authors:** Kai Aoki, Masaki Konno, Katsunari Honda, Takuya Abe, Takeshi Nagata, Masaaki Takehara, Takehito Sugasawa, Kazuhiro Takekoshi, Hajime Ohmori

**Affiliations:** 1Graduate School of Comprehensive Human Sciences, University of Tsukuba, Tsukuba 1-1-1, Tennodai 305-8577, Japan; s1930392@s.tsukuba.ac.jp (K.A.); konno.m.11017325@gmail.com (M.K.); 2School of Physical Education, Faculty of Health and Sport Sciences, University of Tsukuba, Tsukuba 305-8574, Japan; khonda.iba1@gmail.com; 3Zenyaku Hanbai Co., Ltd., Market and Product Development Department. 22-14, Higashiikebukuro 3-Chome, Toshima-Ku, Tokyo 170-0013, Japan; Takuya_Abe@mail.zenyaku.co.jp; 4Zenyaku Kogyo Co., Ltd., Healthcare Research Department of Drug Discovery, R&D Center. 33-7, Ohizumi-machi 2-Chome, Nerima-Ku, Tokyo 178-0062, Japan; Takeshi_Nagata@mail.zenyaku.co.jp; 5Zenyaku Kogyo Co., Ltd., Consumer Product Planning Department. 6-15, Otsuka 5-Chome, Bunkyo-Ku, Tokyo 112-8650, Japan; Masaaki_Takehara@mail.zenyaku.co.jp; 6Faculty of Medicine, University of Tsukuba, 1-1-1 Tennodai, Ibaraki 305-8577, Japan; take0716@krf.biglobe.ne.jp (T.S.); k-takemd@md.tsukuba.ac.jp (K.T.)

**Keywords:** aging, habitual exercise, SAMP8, sarcopenia

## Abstract

Loss of muscle mass and strength are progressing with aging. Exercise is a beneficial method to prevent physical dysfunction, and habitual exercise can improve the muscle quality. Therefore, we evaluated the effects of long-term habitual exercise’s impact on sarcopenia utilizing the senescence-accelerated mice prone8 (SAMP8) model. Notably, 27 w SAMP8 were used in this study. Mice were classified into 28 (28 w) and 44 weeks old. The 44-week group was divided into the sedentary group (44 w) and a group exercising for 16 weeks (44 w + Ex). The 44 w + Ex performed habitual exercise from 28 to 44 weeks. Additionally, grip strength tests were performed with mice aged 28 and 44 weeks. Muscles were harvested and measured muscle weight at 44 w. Gastrocnemius decreased in 44 w, but was unchanged in 44 w + Ex. There was a trend for lower muscle grip strength in the 44 w group, but there was no change in 44 w + Ex. The phosphorylation levels of Akt and p70S6K as a protein synthesis marker were decreased in 44 w. Cytochrome c oxidase subunit IV (CoxIV) mRNA and protein levels decreased in 44 w. These results suggested that long-term habitual exercise attenuates muscle mass and strength decline, possibly through maintenance of muscle protein synthesis and mitochondrial maintenance.

## 1. Introduction

Aging is an inevitable process in animals, including humans, and has various effects on the body. For example, aging affects the functioning of the brain, nerves, and intestines. Skeletal muscle aging includes skeletal muscle loss and muscular weakness, known as sarcopenia [1], Sarcopenia was initially described by Rosenberg [2]. The European Working Group on Sarcopenia in Older People (EWGSOP) provides a definition of age-related sarcopenia and a statement on diagnostic criteria. The EWGSOP recommend using both the loss of skeletal muscle mass and the loss of muscle function (muscle and physical ability) for diagnosis of sarcopenia. This definition is more complex, because muscle strength is not only dependent on muscle mass, but also function [3,4]. In recent years, studies have reported that various forms of exercise have positive effects on preventing sarcopenia. For example, resistance training can stimulate muscle protein synthesis [5,6]. Kryger reported that resistance training by women at an advanced age could increase skeletal muscle mass and muscle strength [7]. More recently, the effects of habitual exercise have also been studied [8,9,10].

Aging affects insulin sensitivity and reduces muscle protein synthesis [11]. Akt is a muscle protein synthesis, related protein, and is upstream protein of muscle protein synthesis. On the other hand, downstream of Akt is 70kDa S6 kinase (p70S6K), which activates muscle protein synthesis. Phosphorylation levels of Akt and p70S6K were previously reported to be low in aged mice and rats [12,13]; therefore, it is suggested that aging has a negative effect on critical signaling pathways for muscle synthesis, and may contribute to sarcopenia. Furthermore, the number and function of mitochondria also decline with age [14,15]. Reduced mitochondrial activity results in reduced ATP synthesis in muscle cells and overall muscle weakness [16,17], for which aerobic exercise has been demonstrated to have a positive effect. Acute aerobic exercise improves the levels of phosphorylated Akt, resulting in enhanced Akt downstream effects on muscle protein synthesis [11]. Some studies reported that 15 months of moderate-intensity training prevented the loss of muscle in the cross sectional area in 18-month-old mice [18], whereas six weeks of moderate-intensity training increased the anabolic function and attenuated the catabolic function in the skeletal muscle of 8-month-old rats. Overall, growing evidence suggests that regular exercise regulates the mitochondrial activity as well as muscle protein metabolism during aging [18]. Short et al. reported that 16 weeks of moderate-to-high intensity regular exercise using a stationary bicycle could increase the mitochondrial activity and cytochrome c oxidase subunit IV (CoxIV) mRNA levels, regardless of age and gender [19]. Moreover, moderate-to-high intensity acute exercise and eight weeks of regular exercise were found to increase the Pgc-1alpha mRNA levels, thereby promoting mitochondrion energy production [20,21]. Positive correlations between muscle ATP synthesis and muscle function have been reported [17], therefore, if mitochondrial functions improve, muscle function may also increase.

In the study of sarcopenia, a validated experimental animal model for addressing aging is still a necessity, as natural aging models are often time consuming due to the need of extended periods of time for accurate assessments. For example, previous aging studies have used F344BN; rats were used as aged models at 30 months old, and as very aged models at 36 months old [13], as well as C57Bl/6 mice, which were used as aged models at 24 months old [12], respectively. In contrast, we adopted the senescence-accelerated mouse (SAM) as an experimental model. SAM represent a multi-factor approach and are an accepted model for accelerated senescence and the study of age-related pathologies [22]. In particular, SAMP8 (the senescence-accelerated mice prone8) was reported as useful in the sarcopenic model [23,24]. Guo et al. reported that the peak of muscle mass and strength in mice was at 7 months of age, followed by a decrease in the next three months [24]. Regarding muscle mass and strength loss, 7-month-old SAMP8 mice correspond to middle-aged regular mice and rats, and 10 months of age is considered to be equivalent to about 24 and 30 months of age in normal mice and rats, respectively [24]. Thus, it is considered that SAMP8 is a valuable model to evaluate sarcopenia in a short period of time compared with the natural aging model [12,13]. However, the underlying molecular mechanism of muscle mass and strength loss in SAMP8 remains unclear. In addition, to our knowledge, there are no studies on the effects of long-term regular exercise programs on this sarcopenia model. Therefore, we hypothesized that muscle protein synthesis and mitochondrial function decline with aging, whereas long-term regular exercise prevents these aging effects. Hence, the aim of this study was to evaluate the effects of long-term habitual exercise on the sarcopenia model.

## 2. Materials and Methods

### 2.1. Animals and Experimental Grouping

The animal experiments in this study were approved by the Animal Care Committee, University of Tsukuba. An overview of the experiments is shown in Figure 1. Male SAMP8 aged 28 weeks (Japan SLC, Hamamatsu, Japan) were kept under conventional housing at 20–26 °C, with a humidity level of 40–60% and a 12-h light/dark cycle. The mice were housed separately in individual cages to avoid fights due to the natural aggressiveness of this mouse strain. Female mice were excluded from this study due to the effects of menstruation and menopause. Mice were freely fed a standard feed (MF, Oriental Yeast, Tokyo, Japan). Mice were divided into a 28-week-old group (28 w) and a 44-week-old group, and split into the sedentary group (44 w) and group undergoing exercise for 16 weeks (44 w + Ex). At the age of 28 weeks, the numbers of the mice were 28 w (*n* = 10), 44 w (*n* = 10) and 44 w + Ex (*n* = 10), but there were mice that died after the initial grip strength measurement and during the study. Therefore, the data used in this study were 28 w (*n* = 8), 44 w (*n* = 7), and 44 w + Ex (*n* = 9).

### 2.2. Exercise Protocol

The mice ran on a rodent treadmill (FVRO, 4E9S-6, Fuji Medical Science, Chiba, Japan). The mice were trained to run for 5 days, followed by training performed at 15 m/min for 30 min a day in light cycle, 5 days per week from 28 to 44 weeks of age. The training protocol was modified from Murase et al. [25]. The back of each treadmill lane contained an electrified grid, which delivered a shock stimulus to the mouse if the mouse stopped running. The mice were monitored during running training to prevent escape.

### 2.3. Grip Strength Test

The grip strength test reflected a muscle strength index of all four limbs and was conducted using a small animal grip measurement device (GMP-100B, MELQUEST, Toyama, Japan). Mice were held in the limb-grip, the tail was pulled backward, and the maximum value of the force applied to release the grip was taken as the measurement value. The measurement was performed five times, and the average value, excluding the maximum value and the minimum value, was taken as the index for each mouse. The same researcher conducted the measurements each time. Measurements were conducted so that the researcher was blind to the individual mouse and measured value to remove bias.

### 2.4. Sampling

At the time of physiological investigation (28 and 44 weeks old; Figure 1), mice were anaesthetized by an intraperitoneal injection of pentobarbitone sodium (Kyoritsu Seiyaku Corporation, Tokyo, Japan). Soleus, plantaris, and gastrocnemius muscles were harvested, following which, mice were euthanized by cervical dislocation. The muscle wet weights were measured at the time of euthanasia and then immediately frozen in liquid nitrogen and stored at −80 °C for later biochemical assays.

### 2.5. Western Blot Analysis

Total proteins were extracted from gastrocnemius muscle with radioimmunoprecipitation assay (RIPA) buffer (1% NP-40, 0.1% SDS, 20 mM Tris-HCl (pH 8.0), 5 mM EDTA, 150 mM NaCl including proteinase inhibitor tablets (cOmplete™ mini, Roche, Basel, Switzerland) phosphatase inhibitor (PhosSTOP™, Roche, Basel, Switzerland)). Lysates were centrifuged at 12,000 g for 15 min at 4 °C. Total protein concentrations for each sample were measured by BCA protein assay kit (Thermo Fisher Scientific, MA, US), and 20 μg/lane of total protein were used for 10% SDS-polyacrylamide gel electrophoresis. In Western blot analysis, the primary antibodies used were rabbit monoclonal Akt antibody (Cell Signaling Technology, #4691T, MA, US, 1:2000), p-Akt antibody (Ser473) (Cell Signaling Technology, #4060T, US, 1:2000), p70S6K antibody (Cell Signaling Technology, #2708T, US, 1:2000), p-p70S6K (Thr389) (Cell Signaling Technology, #9205S, US, 1:2000), CoxIV antibody (Cell Signaling Technology, #4850, US, 1:2000) and OXPHOS antibody (abcam, ab110413, Cambridge, UK, 1:4000). Anti-rabbit IgG, HRP-linked antibody (Cell Signaling Technology, #7074, US, 1:2000) was used as the secondary antibody. A list of antibodies used in this study is shown in Table 1. Signals were detected by chemiluminescence reagent (SignalFireTM Elite ECL Reagent, Cell Signaling Technology, US). Detected images were scanned using a Light-Capture Cooled CCD Camera System (Image Quant LAS-4000, GE Healthcare, UK).

### 2.6. Quantitative RT-PCR Analysis

Total RNA was extracted from the gastrocnemius muscle using Sepasol^®^-RNA I Super G (Nacalai Tesque, Kyoto, Japan). The total RNA concentration was measured with a spectrophotometer (NANO DROP 2000, Thermo Fisher Scientific, MA, USA). Based on the values obtained, the samples were adjusted to 400 ng/μL, with sterile water. After adding 5 × PrimeScript RT Master Mix (RR 036 A, Takara Bio, Shiga, Japan) and RNase-free water to the diluted RNA, reverse transcription was carried out in a thermal cycler (TP 350, Takara Bio) (37 °C, 15 min; 85 °C, 5 s; 4 °C, ∞). After reverse transcription, SYBR Premix Ex Taq II (Tli RNaseH Plus) (Takara Bio) was used for a quantitative polymerase chain reaction (qPCR) technique. Amplification of a reverse transcription polymerase chain reaction (RT-PCR) involved an initial decomposition step at 95 °C for 20 s, decomposition at 95 °C for 30 s, and annealing and extension at 60 °C for 30 s for 40 cycles, using the real-time PCR system 7500 Fast (Applied Biosystems, CA, USA). The mRNA expression of TATA-box binding protein (TBP) was measured as a control. The cycle threshold (Ct) value of the target gene was standardized by the Ct value of the housekeeping gene (ΔCt method). The relative expression level of the target gene was calculated as the relative value compared to the 28-week-old value. The sequences of primers used in this study is shown in Table 2.

### 2.7. Statistical Analysis

Data are shown as the mean ± SEM. Data were subjected to a one-way analysis of variance used to evaluate significance. Tukey’s method was utilized as the post-hoc test analysis. Statistical analyses were performed using SPSS statistics ver. 26 for Mac (SPSS Japan, Tokyo, Japan).

## 3. Results

### 3.1. Body Weight

There were no changes in body weights when the groups were compared. The body weight graphs are displayed in Figure 2.

### 3.2. Skeletal Muscle Mass and Muscle Strength Reduction

To evaluate the effect of aging and habitual exercise on muscle, skeletal muscle mass (Figure 3A), relative skeletal muscle mass and muscle strength were measured (Figure 3B). In the 44 w group, gastrocnemius muscle mass declined with aging and grip strength tended to decrease. In the 44 w + Ex group, there were no significant changes in skeletal muscle and muscle strength. Thus, habitual exercise prevented reductions in skeletal muscle mass and muscle strength associated with aging.

### 3.3. Protein Synthesis Related Protein Expression

To evaluate the effect of aging and habitual exercise on the phosphorylation, the level of Akt (Ser473) and p70S6K were measured (Figure 4). In Akt and p70S6K, 44 w was significant lower compared to both the 28 w and 44 w + Ex groups.

### 3.4. Mitochondrial Functioning Gene and Protein Expression

Habitual exercise increased the expression level of genes related to mitochondrial functioning. To evaluate the effect of aging and habitual exercise on the expression levels of genes related to mitochondrial function, the expression levels of Pgc-1a, CoxIV and Atp5a1 were measured (Figure 5). In the 44 w group, CoxIV mRNA levels declined with aging, and there were no changes in Pgc-1a and Atp5a1. In the 44 w + Ex group, there were no changes in CoxIV and Atp5a1. CoxIV gene expression was changed in the 44 w mice, so we examined the protein expression in COXIV. COXIV gene expression in the 44 w mice was significantly lower than both the 28 w and 44 w + Ex groups.

## 4. Discussion

The loss of skeletal muscle mass and strength, collectively known as sarcopenia, is a significant problem in terms of quality of life, health, and decreased life expectancy. Muscle aging studies and muscle aging animal models are required, since human life is of considerable length, making longitudinal examinations very difficult. In this study, we used the SAMP8 for a sarcopenia model animal. First, we examined muscle mass and muscle strength changes following a habitual exercise program. Guo et al. reported that SAMP8 muscle mass peaked at 7 months old and muscle strength decline began at 8 months old; therefore, we used 28 weeks old (7 months old) SAMP8 [24]. Relative gastrocnemius muscle mass decreased with aging (Figure 3B). Although the absolute weights tended to be significant, there was no significant difference. This point was different from previous studies. Liu et al. reported that fat accumulates in SAMP8′s muscles with aging [26]. Hence, ectopic fat may have accumulated in this study. Similarly, muscle strength decline did not show significant differences; however, the 44 w group showed a significant trend. Takeshita et al. reported that the normal grip strength test performed in parallel to the ground can be subjected to errors, because it depends on various factors such as the examiner assessment accuracy and the motivation of the rodent to keep gripping the bar [27]. We performed the normal grip strength test; thus, it is possible that an error occurred, thereby contributing to the inability to obtain a clear result. Additionally, we did not observe clear changes in muscle weight and strength, which was in contrast to the findings of previous studies. This could be explained by differences in the housing environment of the animals, which could have contributed to differences in physical activity, among other effects. Therefore, when evaluating SAMP8 as a model of sarcopenia, it is desirable to measure the size of muscle fibers by tissue sectioning and the direct muscle tension exerted ex vivo or through other methods. Nevertheless, it is useful to be able to evaluate the effects of aging on muscles in a short period of time, so SAMP8 is a useful model animal for sarcopenia.

To evaluate why sarcopenia was attenuated by long-term, habitual exercise, muscle protein synthesis-related proteins and genes related to mitochondrial function were investigated.

Muscle atrophy is a multifactorial process, yet a decline in muscle protein synthesis contributes [14]; therefore, phosphorylation levels of Akt and p70S6K, which are related to muscle protein synthesis, were measured. In this study, the phosphorylation level of proteins showed a decline with aging. Therefore, muscle atrophy at this age of SAMP8 may be primarily due to declined muscle protein synthesis. However, some contradictory results were observed between muscle protein synthesis, related protein expression, and muscle mass. As mentioned above, ectopic fat is reported to accumulate with aging in the muscle of SAMP8 mice. We cannot state that fat may have contributed to these contradictory findings, as we have not measured the amount of triglycerides in the muscle; however, we believe that the actual muscle mass was reduced at 44 weeks.

Long-term, habitual exercise can improve mitochondrial function and increase mitochondria content, therefore, we examined the genes and proteins related to mitochondrial function. During aging, mitochondrial function declines and this causes muscle strength to also decline. In this study, the mRNA and protein expression levels of CoxIV, which is a mitochondrial component, were decreased at 44 w (Figure 5). In this sarcopenia model, aging induced a decrease in mitochondrial activity and/or a decrease in mitochondrial content; however, these effects were attenuated by exercise, which contributed to maintaining muscle strength. Direct muscle tension measurements should be addressed further, which may yield clearer results.

Regarding exercise intensity, 18–19 m/min is considered to be the lactate threshold (LT) in mice [20]; therefore, exercise below the LT would be categorized as low-to-moderate intensity. In the current study, the exercise intensity reached this level (i.e., 15 m/min). Although some results on muscle mass, strength, and related protein expression pattern were somewhat inconsistent with previous reports, this is the first time that the effect of long-term exercise was evaluated in SAMP8 as a sarcopenia model. Therefore, the present study provides valuable findings that suggest that regular low-intensity exercise may help to maintain muscle synthesis, mass, and strength, as well as mitochondrial function. Generally, it is difficult for elderly people to exercise with high intensity. However, our results suggest that even low-intensity exercise, which is safe and executable for the elderly, could attenuate aging-associated muscle mass and strength decline through the maintenance of mitochondrial function and muscle protein synthesis. Of course, it is highly conceivable that the muscle mass and strength, and the expression patterns of genes and proteins related to them, will be altered by the change of exercise intensity, time, and period. Therefore, in order to evaluate further the effects of regular exercise on sarcopenia, additional studies in SAMP8 mice are warranted.

Despite these findings, the mitochondrial genome level and mitochondria content were not measured in this study. In addition, muscle fiber type and fiber cross sectional area could not be examined using muscle sections, creating a limitation which must be addressed in future study.

## 5. Conclusions

The effects of long-term, habitual exercise in SAMP8 mice, which were utilized as an aging model to investigate the effects of the sarcopenia model, was evaluated in this study. Long-term, habitual exercise preserved muscle mass and muscle strength. The mechanisms underlying this phenomenon could be preservation of the muscle protein synthesis and mitochondrial function.

## Figures and Tables

**Figure 1 geriatrics-05-00048-f001:**
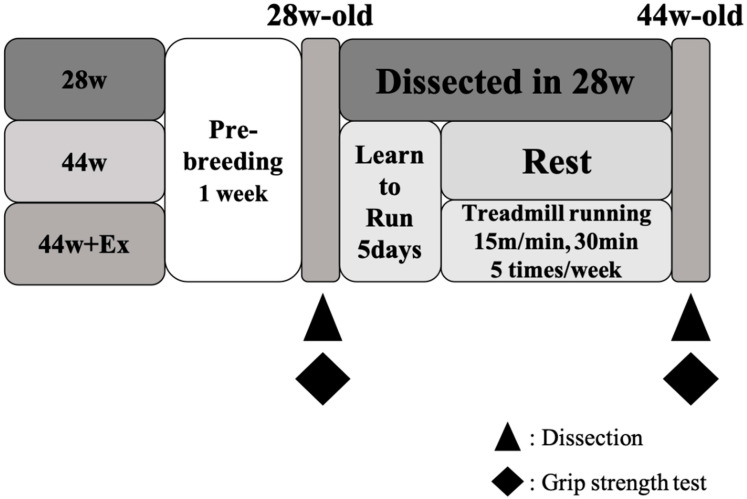
Overview of this study.

**Figure 2 geriatrics-05-00048-f002:**
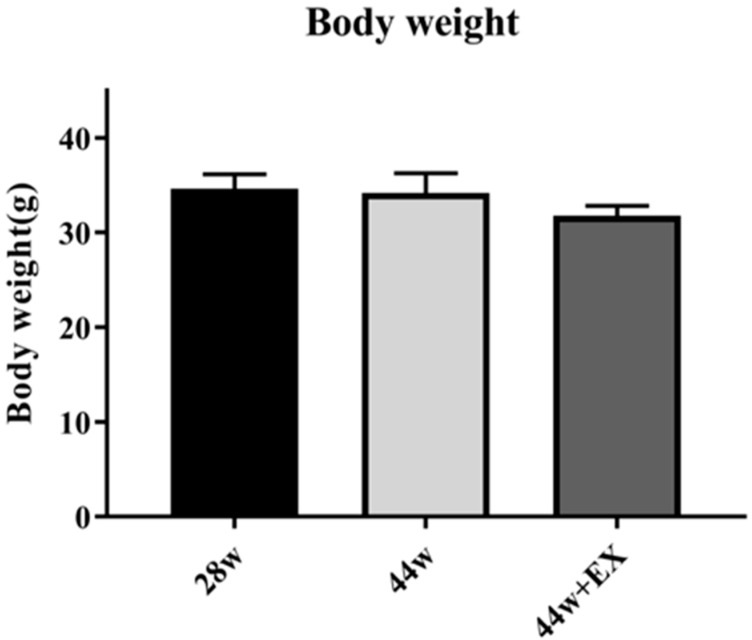
Body weight in each group. Body weights were measured before euthanasia at each time point. Values are means ± SE (*n* = 7–9).

**Figure 3 geriatrics-05-00048-f003:**
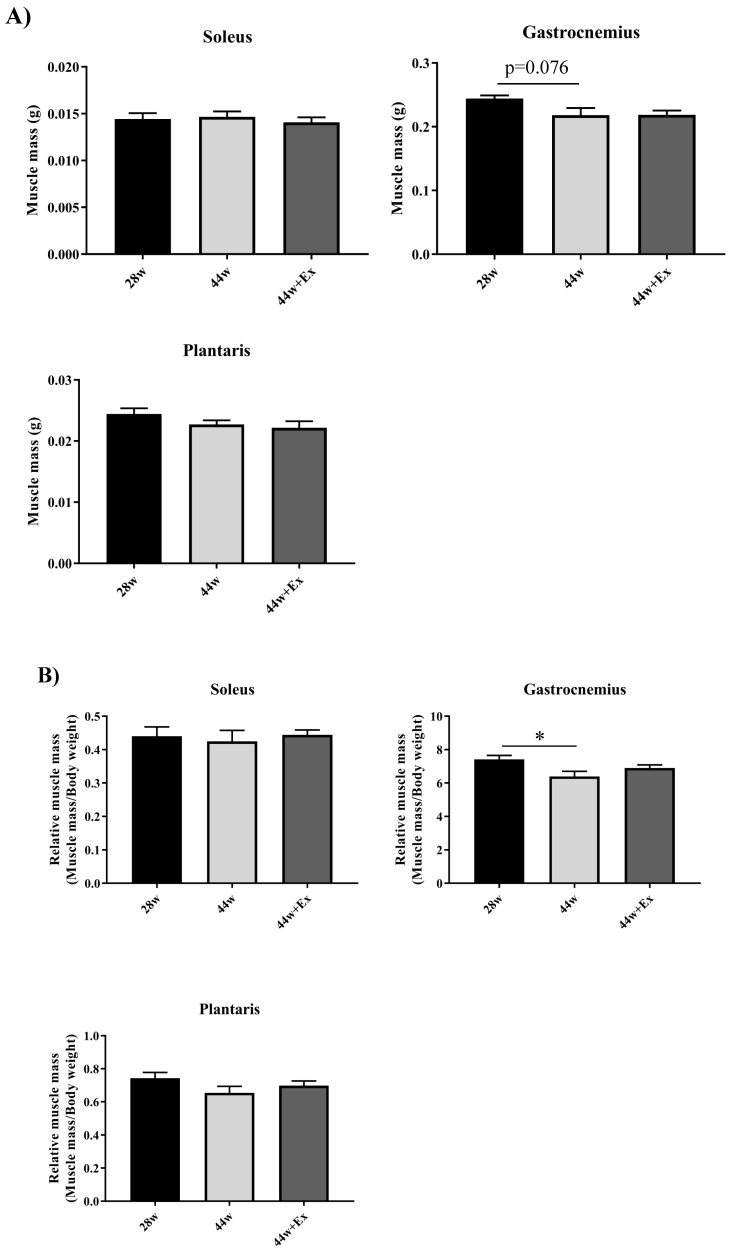
Muscle weight and Grip strength. (**A**) Muscle mass in each group. (**B**) Relative muscle mass. (**C**) Relative Grip strength. Muscle weights were measured at each time point. The grip strength reflected a muscle strength index of all four limbs and was conducted using a small animal grip measurement device. Values are means ± SE (*n* = 7–9). Single asterisk indicates significant differences (*p* < 0.05).

**Figure 4 geriatrics-05-00048-f004:**
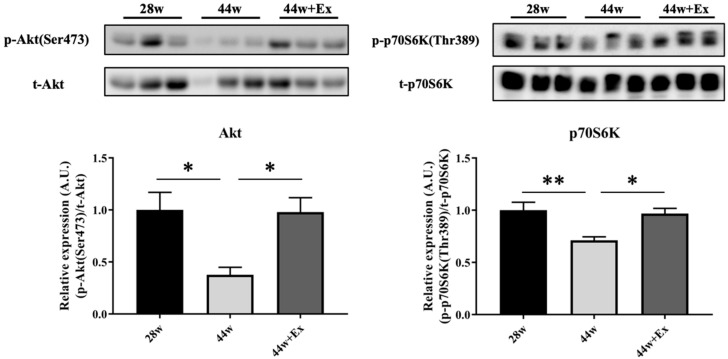
The phosphorylation level of Akt (Ser473) and p70S6K in gastrocnemius muscle. Total protein was extracted from gastrocnemius muscle and both proteins were analyzed by Western Blot analysis. Akt: Protein kinase B, p70S6K: 70-kDa ribosomal protein S6 kinase. 28 week old (28 w), 44 week old (44 w) and exercise (44 w + Ex). Values are means ± SE (*n* = 6). Double asterisks indicate significant differences (*p* < 0.01). Single asterisk indicates significant differences (*p* < 0.05).

**Figure 5 geriatrics-05-00048-f005:**
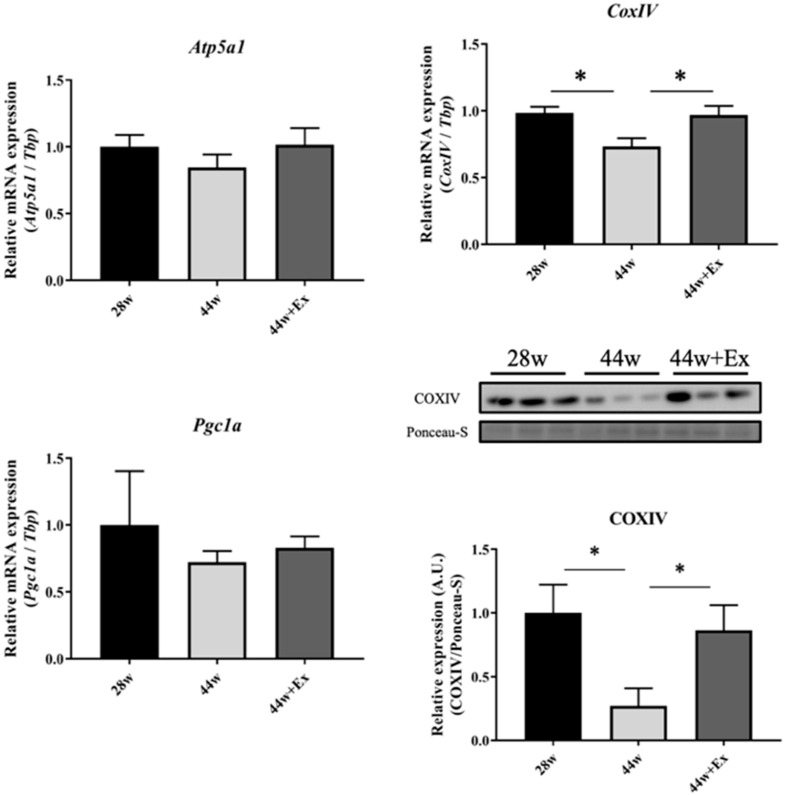
Genes and protein expressions related to mitochondrial function in gastrocnemius muscle. Genes expression level, related to mitochondrial function. Values are means ± SE (*n* = 7–9). Proteins expression level, related to mitochondrial function. Values are means ± SE (*n* = 6). 28 week old (28 w), 44 week old (44 w) and exercise (44 w + Ex). Single asterisk indicates significant differences (*p* < 0.05).

**Table 1 geriatrics-05-00048-t001:** List of antibodies.

Product Code	Protein Name	Spiecies	Dilution
CST #4691T	Akt	Rabbit	1:2000
CST #4060T	p-Akt (Ser473)	Rabbit	1:2000
CST #2708T	p70S6K	Rabbit	1:2000
CST #9205S	p-p70S6K (Thr389)	Rabbit	1:2000
CST #4850	CoxIV	Rabbit	1:2000
CST #7074	Anti-rabbit IgG, HRP-linked Antibody	Rabbit	1:2000

**Table 2 geriatrics-05-00048-t002:** List of primers.

Accession No.	Gene Name	Primer Sequence	bp
NM_008904.2	PPAR gamma coactivator 1-alpha	5′-GCGAACCTTAAGTGTGGAACTC-3′	93
	(PGC-1α)	5′-GCCTTGAAAGGGTTATCTTGG-3′	
NM_001293559.1	Cytochrome c oxidase subunit 4	5′-TGATGTGGCCCATGTCAC-3′	121
	(CoxV)	5′-GTTAAACTGGATGCGGTACAACT-3′	
NM_007505.2	mitochondrial H+-ATP synthase alpha subunit	5′-TCCATGCCTCTAACACTCGAC-3′	122
	(Atp5a1)	5′-GCTTAACACACGCCCAGTCT-3′	
NM_013684.3	TATA box binding protein	5′-CCAATGACTCCTATGACCCCTA-3′	104
	(Tbp)	5′-CAGCCAAGATTCACGGTAGAT-3′	

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
