# Peer review of "Habitual Aerobic Exercise Diminishes the Effects of Sarcopenia in Senescence-Accelerated Mice Prone8 Model"

_geriatrics, 2020, doi:10.3390/geriatrics5030048_

Round 1

Reviewer 1 Report

Version:1.0 StartHTML:0000000119 EndHTML:0000004968 StartFragment:0000000909 EndFragment:0000004954

This study evaluated the effects of a long-term habitual exercise on the senescence-accelerated 26 mice prone8 (SAMP8). It is thought to be a valuable study that suggested the possibility of preventing muscle mass loss through exercise.

However, the following important problems need to be supplemented.

1.   In the introduction, data related to previous studies on the effects of longterm habitual exercise need to be supplemented.

2.   In the discussion, an explanation of the results that did not show a statistically significant difference in grip strength is required.

3.   In the discussion, it is necessary to explain why the protein expression patterns and muscle mass changes related to muscle synthesis are inconsistent.

4.   It is necessary to consider that the content of exercise applied in this study and the effect on protein expression and mitochondrial biosynthesis related to muscle synthesis may appear differently.

Reviewer 2 Report

Reviewer Recommendation and Comments for Manuscript #: Geriatrics-899884:

Habitual exercise attenuates the aging associated muscle atrophy and muscle strength decline in SAMP8”

GENERAL:

            The current manuscript is a very interesting and very important investigation that probes the question owing to sarcopenia and the effectiveness of an exercise intervention to attenuate the sarcopenic trajectory. The authors utilize a fundamental senescence-accelerated mice prone8 (SAMP8) model for investigation. These broad areas of research have been and continue to be very active areas, and both necessitate more investigation, especially for informing translational research. This manuscript was a pleasure and a challenge to critically review. Even though the results are succinct, they do not necessarily support an attenuation of sarcopenia across the multiple physiological- and performance-level outcomes. Indeed, these data, even though direct, would/could be detailed in a more relative fashion if the authors would provide a more elaborate explanation for the use of the SAMP8 aging model; this would be helpful if laid-out to the reader in the Introduction. Given the ages the authors employed in the current study, if the chronological ages of the SAMP8 aging model is of extreme biological/physiological age then the study’s findings have enhanced value. Further, the authors do not provide an explicit hypothesis(es) for testing purposes, which should be included. These concepts should be considered for modification in the current manuscript, for the team’s current research is a very important aspect of contemporary gerontological/aging research involving the efficacy of interventions to offset these deleterious outcomes. Even though there are multiple points that should be addressed concerning the existing manuscript, the authors should be able to address these points for full consideration.

TITLE

Given the specific significance reported by the authors related to muscle mass only (not performance – i.e. muscle grip strength), it is recommended that the authors alter the title to be more encompassing. An example would be: “Habitual aerobic exercise diminishes the effects of sarcopenia in senescence-accelerated mice prone8 model”

ABSTRACT

Page 1, Lines 25: Change “disfunction” to “dysfunction”

Page 1, Line 25: Add “comma” following “dysfunction”

Page 1, Line 25: Add “can” following “exercise”

Page 1, Line 26: Delete “a” prior to “long-term”

Page 1, Lines 26-27: Change “…exercise on the senescence-accelerated mice prone8 (SAMP8).” to “…exercise’s impact on sarcopenia utilizing the senescence-accelerated mice prone8 (SAMP8) model.”

Page 1, Lines 30-31: Delete “Mice were dissected and collected”; start the sentence with “Muscles were harvested and measured…”

Page1, Line 32: Change “Grip strength in 44w was lower trend…” to “There was a trend for lower muscle grip strength in the 44wk group…”

Page 1, Line 35: Change “attenuated” to “attenuates”

Page 1, Line 36: Change “…decline through improving muscle protein synthesis and mitochondrial function” to “…decline; possibly, through maintenance of muscle protein synthesis and mitochondrial maintenance.”

Page 1, Lines 36-37: Delete final sentence

INTRODUCTION

Page 2, Line 48: Change “in” to “on”

Page 2, Line 51: Change “one of the” to “a”

Page 2, Line 51: Add “hyphen” between “synthesis-related”

Page 2, Line 52: Change “hands” to “hand”

Page 2, Lines 53-54: Change “…Akt is decreasing in the old mice has been reported” to “…Akt was previously reported to be decreased in aged mice.”

Page 2, Line 54: Clarify the sentence “As a result…” this is not coherent.

Page 2, Line 55: Change “activate the” to “induces downstream effects on”

Page 2, Line 61: Add “semicolon” after “[16]

Page 2, Line 62: Change “…aging model animals are required” to “…a validated model of aging utilizing old animals is a necessity”

Page 2, Line 62: Change “…a lot of time…” to “…extended periods of time…”

Page 2, Line 65: Change “sarcopenia” to “sarcopenic”

Page 2, Line 66: Change “sarcopenia” to “sarcopenic”

Page 2: The authors do provide the central aim of the current study, however the authors must explicitly state and provide the/a specific hypothesis(es) that was tested.

METHODS

2.1 Animals and Experimental Grouping:

Overall, will the authors elaborate on the relative age of a 28w and 44w SAMP8 mouse versus other mouse strains (e.g. C57B6)? This is important, not only to the reviewer but to the audience/readership, because a 28w (i.e. 7m) mouse is not sarcopenic, while a 44w mouse may only beginning to show implicit metrics of sarcopenia. This/these point(s) should be included in the Introduction.

Page 2, Line 73: The authors should elaborate upon the statement “…were kept under conventional in individually…”. “Under conventional” What? “In individually” What?

Page 2, Line 76: Delete “the” prior to “28 weeks” and “the” prior to “44 weeks”

2.2 Exercise Protocol:

Page 3, Line 84: Change “…ran on a treadmill for small animals…” to “…ran on a rodent treadmill…”

Page 3, Line 85: Change “The mice learned to run…” to “The mice were trained to run…”

Page 3, Line 86: Change “Training protocol were modified…” to “The training protocol was modified…”

Page 3, Line 88: Change “…to mice if mice are resting on grid.” to “…to the mouse if the mouse stopped running.”

Page 3, Lines 89-90: Delete final sentence

2.3 Sampling

Page 3, Line 105: Delete “muscles” following “Soleus” and following “plantaris”

2.4 Western Blot Analysis

Page 4, Line 116: Add “comma” following “…Scientific, MA, US),

Page 4, Line 123: Change “was” to “were”

Page 4, Line 124: Change “antibody.” to “antibodies.”

2.6 Statistical Analysis

Page 5, Line 150: Change “Post hoc test was used Tukey’s method.” To “Tukey’s method was utilized as the Post-hoc test analysis.”

RESULTS

3.3. Protein Synthesis Related Protein Expression

Page 7, Line 179: Change “…level of Akt…” to “…level of pAkt(Ser473)…”

Page 7, Line 193-194: Change “…genes expression was change 44w so we examined…” to “…gene expression was change in the 44w mice, so we examined…”

Page 7, Line 194: Change “In COXIV 44w was…” to “COXIV gene expression in the 44w mice was…”

Page 4, Line 187: Change “Besides...” to “In addition…”

DISCUSSION

Page 8, Line 217: Change “Muscle aging, known as sarcopenia, is significant…” to “The loss of skeletal muscle mass and strength, collectively known as sarcopenia, is a significant…”

Page 8, Line 219: Add “longitudinal” prior to “examinations”

Page 8, Line 229: Add “may have” following “environment”

Page 8, Line 235: Change “… is caused by a decline in muscle protein synthesis [14], therefore, phosphorylation…” to “…is a multifactorial process, yet a decline in muscle protein synthesis contributes [14]; therefore, phosphorylation…”

Page 8, Lines 237-238: Change “…showed decling with aging and exercise suppressed its decline.” to “…showed a decline with aging, while exercise attenuated this decline.”

Page 8, Lines 240-242: Change “Previous study referred that 18-19 m/min is lactate threshold (LT) in mice [19]. Exercise below LT was low to moderate intensity, and it is considered that this study was low to moderate intensity of 15 m/min.” to “Given that 18-19 m/min is reported to be considered the lactate threshold (LT) in mice [19], exercise below the LT would be categorized as low to moderate intensity, and the current study’s intensity reached this level (i.e. 15 m/min).”

Page 8, Lines 242-243: Delete final sentence

Page 9, Lines 246-247: Delete sentence two

Page 9, Line 248: Change “level” to “levels”

Page 9, Line 249: Change “marker” to “component” and Change “in” to “at’

Page 9, Line 251: Change “had inhibitory” to “attenuated this”

Page 9, Line 252: Change “…even though low intensity exercise that is safety…” to “even utilizing low-intensity exercise that is safe…”

Page 9, Line 253: Add “comma” following “decline”

CONCLUSIONS

Page 9, Line 259: Change “…mice as a sarcopenia model were…” to “…mice, which were utilized as an aging model to investigate the effects of sarcopenia…”

FIGURES/FIGURE LEGENDS

Figure 2:

Page 5, Line 163: Delete second sentence

Page 5, Line 164: Change “…dissection in each time points.” to “…euthanasia at each time point.”

Figure 3:

Page 6: Figure 3 should be presented as three separate sub-figures: A) Absolute Muscle Mass, B) Relative Muscle Mass, and C) Muscle Grip Strength.

Page 6, Line 173: Change “in” to “at” and Change “points” to “point”

Page 6, Line 176: Delete sentence referring to ages/groups of animals (i.e. 28week-old (28w), etc,)

Figure 4:

            Page 7, Line 182: Change “…level of Akt…” to “…level of pAkt(Ser473)…”

Figure 5:

            Page 8, Line 198: Add “hyphen” between “level-related”

            Page 8, Lines 199-205: Delete sentences three and four

            Page 8, Lines 207-211: Delete sentences seven and eight

REFERENCES

Round 2

Reviewer 1 Report

The author has modified it appropriately. Thank you.

Reviewer 2 Report

The authors should be commended for the modifications made to the manuscript; it is now clearly and succinctly presented. The authors have satisfied all concerns/comments raised by this reviewer.